# Wild Plants Used as Herbs and Spices in Italy: An Ethnobotanical Review

**DOI:** 10.3390/plants10030563

**Published:** 2021-03-16

**Authors:** Riccardo Motti

**Affiliations:** Department of Agricultural Sciences, University of Naples Federico II Via Università, 100 80055 Portici, NA, Italy; motti@unina.it

**Keywords:** ethnobotany, food plants, spices, herbs, Italy

## Abstract

Wild edible plants are an essential component of people’s diets in the Mediterranean basin. In Italy, ethnobotanical surveys have received increasing attention in the past two centuries, with some of these studies focusing on wild edible plants. In this regard, the literature in Italy lacks the coverage of some major issues focusing on plants used as herbs and spices. I searched national journals for articles on the use of wild food plants in Italy, published from 1963 to 2020. Aims of the present review were to document plant lore regarding wild herbs and spices in Italy, identify the wild plants most frequently used as spices, analyze the distribution of wild herbs and spices used at a national scale, and finally, to describe the most common phytochemical compounds present in wild plant species. Based on the 34 studies reviewed, I documented 78 wild taxa as being used in Italy as herbs or spices. The studies I included in this systematic review demonstrate that wild species used as herbs and spices enrich Italian folk cuisine and can represent an important resource for profitable, integrated local small-scale activities.

## 1. Introduction

In recent decades, detailed ethnobotanical studies have revealed the widespread use of wild plants in the Mediterranean basin [1,2,3,4]. Ethnobotany is a multidisciplinary investigation of interrelations between people and plants [5], and plays a key role in ascertaining the various plant species used in traditional cuisine. Indeed, ethnobotany of food plants is a fairly well-developed research field in several geographical areas and social communities [6,7,8]. Moreover, ethnobotanical studies concerning food plants offer novel ways to analyze and preserve traditional knowledge and agrobiodiversity in the Mediterranean area [3]. 

Herbs and spices produced from aromatic plants are largely used to enhance food taste and palatability. Sometimes used as synonyms, the distinction between the two terms (herbs and spices) could be summarized as follows: herbs are types of plants whose leaves are used in cooking to give flavor to particular dishes, while a spice is defined as any of the various aromatic products obtained from plants in the form of powder or seeds or other plant parts and used to add taste to food (e.g., [9,10,11,12]). According to Van der Veen and Morales [13] neither definition fully conveys the array of plants nor the range of purposes for which such plants are used. A further lexical complication is that, in the ethnobotanical literature, there are many terms to indicate the use of plants in cooking, e.g., aromatic, aromatizer, condiment, flavoring, and spice. In the present review, I refer to the wild species (the whole plant or parts of it) used for flavoring various dishes in the Italian folk tradition. In this context, I use the term “wild” to refer to non-cultivated or naturally occurring plants gathered in the field, although sometimes for convenience of use, some species are grown or deliberately tolerated in home gardens [14,15]. In some cases, food plants are also eaten for their health-giving properties and many species are commonly used as herbal medicines in folk phytotherapy for the treatment of ailments [16,17].

In Italy, ethnobotanical surveys have received increasing attention in recent decades, seeking to investigate the traditional uses of plants and their products (e.g., [18,19,20]). Some of these studies focus on wild edible plants (e.g., [21,22]), while others infer the use of wild plants as food through more general studies of ethnobotany (e.g., [23,24]). In this regard, the literature in Italy lacks the coverage of some major issues focusing on plants used as spices.

In this context, the aim of the present work is to review and highlight the use of wild plants as traditional herbs and spices in folk cuisine in Italy. 

The specific aims of this study were: (a) to document plant lore regarding wild herbs and spices in Italy; (b) to identify the wild plants most frequently used as spices; (c) to analyze the distribution of wild herbs and spices used at a national scale; (d) to describe the most common phytochemical compounds present in wild plant species.

## 2. Results

### General Data

Based on the 34 studies providing adequate and relevant data, I documented 78 wild taxa as being used in Italy as herbs or spices (Table 1). The plant species belong to 19 families and 49 genera. Lamiaceae (38.8%) is the most frequently cited family, followed by Amaryllidaceae (12.5%), Apiaceae (11.3%), Asteraceae (7.5%) and Rosaceae (6.3%). The most species-rich genus is Allium (9 species), followed by Mentha (8), Thymus (6) and Salvia (5). Leaves (63.0%) were the most frequently used plant parts, followed by fruits (10.9%), and flowers (8.7%). The remaining parts (including bulbs, seeds and roots) accounted for 17.4% overall.

As shown in Figure 1a, from the analyses carried out at a regional scale, I found that Tuscany has the largest number of species used in a single region (24), followed by Campania (23), Abruzzo and Lombardy (22 each). The Ethnobotanicity Index values are significantly lower than those of other Italian regions (range 5.4–11%) or Iberian Peninsula (range 8.8–27.9%) that are calculated on medicinal, cosmetic, veterinary, and food species [17,52,53]. Although it is not possible to make a comparison with the same use category, results suggested that the knowledge of wild plants used as flavoring is still consolidated in the above-mentioned regions that are also among the most species-rich in Italy [54]. (Figure 1b). Papers containing reports of wild plants specifically used for flavoring are reported in the Appendix A.

## 3. Botany and Phytochemistry

In Figure 2, I present a summary of the sixteen most commonly cited taxa Italy. For each species, I discuss its life form, chorology and phytochemical profiles.

Fennel (*Foeniculum vulgare* Mill.) is a hardy, perennial herb native to Mediterranean coasts and widely naturalized in many parts of the world. Fennel seed is a rich source of volatile oil, with its main compounds being fenchone and trans-anethole. Other components of the essential oil are limonene, camphene, estragole and α-pinene [55,56]. The main constituents of the essential oils extracted from its leaves are trans-anethole, estragole, fenchone, and α-phellandrene; minor constituents are limonene, neophytadiene and phytol [57,58].

Bay laurel (*Laurus nobilis* L.) is an evergreen tree or large shrub native to the Mediterranean region. The aromatic leaves are rich in essential oils whose main components are: 1,8-Cineole, sabinene, α and β-pinene, α-terpinylacetate and linalool [59,60].

Rosemary (*Rosmarinus officinalis* L.) is an evergreen sclerophyllous shrub native to the Mediterranean basin. The main components of the essential oil are 1,8-Cineole, α and β-pinene, camphor, camphene and *β*-pinene [61]. Rosmarinic acid is an ester of caffeic acid and 3,4-dihydroxyphenyllactic acid, with a number of interesting biological activities (e.g., antiviral, antibacterial, anti-inflammatory and antioxidant) and is widely occurring in the Lamiaceae family.

The common juniper (*Juniperus communis* L.) is a multistemmed shrub or small tree, whose seed cones are usually called berries. This is the most widespread conifer in the world, native to temperate Eurasia, North America and northern Mexico, occupying an extraordinary range of habitats [62]. Juniper berry oil largely consists of monoterpene hydrocarbons such as 𝛽-pinene, 𝛼-pinene, sabinene, myrcene, and limonene [63]. 

Oregano (*Origanum vulgare* L. subsp. *vulgare*) is a perennial herb native to the temperate regions of Europe and Asia. The main constituents of the essential oil in its leaves are carvacrol, p-cymene, c-terpinene, limonene, terpinene, ocimene, caryophyllene, β-bisabolene, linalool, and 4-terpineol [64]. Oregano is added for its slightly bitter flavor to poultry, fish, and other dishes. *O. vulgare* L. subsp. *viridulum* (Martrin-Donos) Nyman is also reported as a wild spice.

Sage (*Salvia officinalis* L.) is an evergreen subshrub native to the Mediterranean basin. The principal components in sage oil are 1,8-cineole, camphor, α-thujone, β-thujone, borneol, viridiflorol, caryophyllene and cineole [65,66]. Other Salvia species (*S. glutinosa* L., *S. pratensis* L., *S. sclarea* L., *S. verbenaca* L.) are also used. 

Thyme (*Thymus longicaulis* C. Presl) is an evergreen subshrub native to southern Europe. Thyme essential oil consists of highly variable amounts of phenols, monoterpene hydrocarbons, and alcohols. Thymol is normally the main phenolic component followed by carvacrol [67]. The variability of the chemical components of Thymus species depends on several parameters including climatic, seasonal, and geographic conditions [68]. For *Thymbra capitata* (L.) Cav., *Thymus polytrichus* Kern. ex Borbás, *T. pulegioides* L. and *T. spinulosus* Ten are also reported to have the same uses of *Th. longicaulis*.

Lesser calamint (*Clinopodium nepeta* (L.) Kuntze subsp. nepeta) is an erect herbaceous perennial species, sometimes woody at the base, native to southern Europe. Its essential oil is rich in menthone, pulegone, piperitone, neomenthol, menthol, and limonene [69,70]. 

Winter savory (*Satureja montana* L.) is a perennial, semi-evergreen subshrub, native to warm temperate regions of southern Europe and Africa. The volatile fraction of winter savory essential oil is mainly characterized by oxygenated monoterpenes like thymol and carvacrol [71]; other important compounds are the monoterpenic hydrocarbons p-cymene and γ-terpinene [72].

Chives (*Allium schoenoprasum* L.) is a herb native to cold and temperate areas of Europe, Asia and North America; in Italy, it grows on mountains in central and northern regions. Green leaves of chives have sulfur compounds like 2-methyl-2-butenal, 2-methyl-2-pentenal, methyl-propyl disulfide and dipropyl disulfide. The major thiosulfinate compounds from chives are n-propyl groups, methyl and 1-propenyl groups [73]. 

Lemon balm (*Melissa officinalis* L.) is a perennial herbaceous plant native to central Europe, the Mediterranean Basin, and Central Asia, but now naturalized in the Americas and elsewhere [74] (WCSP 2020). The main components of the essential oil are citronellal, citral (citronellol, linalool) and geranial. In addition, this oil contains three terpinene, rosmarinic acid, and flavanol glycoside acids in low ratio [75,76]. 

Capers (*Capparis spinosa* L.) is a deciduous shrub native to Europe and Asia. The floral buds are harvested still closed in spring and summer and usually processed in brine. Cinnamaldehyde and benzaldehyde are the major constituents of the flavor profile of capers; of sulfur compounds, methyl-isothiocyanate is the main compound, followed by benzyl-isothiocyanate [77]. Rutin and kaempferol are the most abundant flavonol glucosides [78,79]. 

Garlic (*Allium triquetrum* L.; *A. ursinum* L.) is a bulb-forming herbaceous perennial plant, reported in the present review as a herb to flavor several dishes. Thiosulfinates are responsible for its characteristic pungent aroma and taste [80]. When fresh garlic is chopped or crushed, the enzyme alliinase converts alliin into allicin, which is responsible for the aroma of fresh garlic [81]. The typical aroma of cooked garlic is due to allyl methyl trisulfide [82]. Garlic is also considered a functional food and is widely used for its antibacterial, hypoglycemic, hypotensive and hypocholesterolemic properties [83]. It is used in folk phytotherapy also as a galactofuge and anthelminthic [84,85]. Other wild garlics reported as herbs in the present review are: *A. lusitanicum* Lam., *A. neapolitanum* Cirillo, *A. roseum* L., *A. subhirsutum* L. and *A. vineale* L.

## 4. Discussion

The use of some species is linked to their presence in the various regional territories. *Gymnadenia rhellicani* (Teppner and E.Klein) Teppner and E.Klein, for example, is an orchid confined to the Alps, at altitudes of 1000–2800 m [86]. This species is reported as a sweet flavoring only in two papers concerning alpine ethnobotany in Lombardy [17,23]. Similar considerations can be made for *Pinus mugo* Turra, a species growing at high altitudes and reported in the same papers. *Pimpinella anisum* L. is a casual alien distributed in a few regions, whose use is reported only for Abruzzo and Lombardy [16,23], while the range of *P. anisoides* V. Brig. is in central-southern Italy and its use is reported only in Calabria [38,49,50]. *Allium schoenoprasum* L. is not spontaneous in southern and insular regions and, although it is also frequently grown in gardens throughout Italy, its use is reported only for central and northern regions.

Many herbs and spices cited in the present review are also reported for their positive influence on health, especially for gastrointestinal disorders and more frequently as a digestive or appetizer (see Table 1). Many of these plants (e.g., *Thymus* spp., *Mentha* spp., *Salvia* spp., *Foeniculum vulgare*) are well known for their ability to stimulate the excretion of digestive enzymes and their carminative properties [66,68,87]. Many herbs and spices (e.g., *Rosmarinus officinalis*, *Foeniculum vulgare*, *Allium* spp.) have other nutraceutical properties and in particular are rich in antioxidant compounds. Many studies (e.g., [88,89]) have highlighted that a dietary antioxidant intake has a protective effect against free radical-related pathologies, such as cardiovascular diseases, diabetes, cancer and neurodegenerative diseases. Recent studies have highlighted that the protective effect of nutraceuticals is linked to the association of several phytochemical molecules at low concentrations, as it occurs naturally in the diet [90]. These plants can, therefore, be identified as functional foods (foods that have beneficial effects on one or more functions of the human organism that go beyond their mere nutritional properties [91,92]), and are consumed because they have a positive effect on health. Moreover, many of the food plants mentioned above are also reported for their herbal uses and are consumed as tea or used for topical applications.

Wild herbs and spices also play an important role in traditional gastronomy because they are used in the recipes of many local dishes, and in a certain way, they contribute to the cultural identity of some geographical areas. According to Jordana [93], in order to be traditional, a product must be linked to a territory and it must also be part of a set of traditions, which will necessarily ensure its continuity over time. Such wild species are, indeed, related to the preservation of family or local traditions and, as pointed out by Luczaj et al. [2] and Pardo-de-Santayana et al. [15], they could also be considered a way to diversify the daily diet.

According to Pieroni et al. [14], the potential of wild plants should be further explored for the possible economic opportunities that could be generated for local gatherers and communities. The diversification of production using such resources could be a socio-economically sustainable activity in areas with non-optimal farming conditions by contributing to population stabilization in rural areas [94]. Herbs are, therefore, also an opportunity to develop a healthy diet that combines gastronomy, health and sustainability.

In conclusion, the studies we included in this systematic review demonstrate that wild species used as herbs and spices enrich Italian folk cuisine and can represent an important resource for profitable, integrated local small-scale activities. Wild food species contribute to local food systems and to the local gastronomy, playing an important role in the economy of small communities.

The role of ethnobotanical studies is to avoid the loss of traditional knowledge concerning the use of food plants and, at the same time, provide the basis for developing new drugs from phytochemical and biochemical research. 

In this regard, new field investigations aimed at specific knowledge of wild herbs and spices are desirable in Italy.

## 5. Materials and Methods

### 5.1. Geographical Context

Italy comprises some of the world’s most varied and scenic landscapes [95] and has an excursion of about 12 degrees of latitude (between 35° and 47° N). According to the Köppen climate classification [96], Italy is divided into ten types of climate [97]. The country, therefore, has an extreme variability of environments, ranging from coastal areas to the high altitudes of the Apennines and Alps. The whole Italian flora comprises 8195 native taxa, which is the highest number in Europe [98].

### 5.2. Data Collection

I searched both national and international journals for articles on the use of wild food plants in Italy from 1950 to 2020 and the first relevant publications date back to 1963. Publications were collected from online versions of the Science Citation Index, Elsevier Journal Finder, ISI web of knowledge, Scopus, and Google Scholar using the key words: ethnobotany, wild food plants, Italy. Further articles and books were gathered from previously collected papers. The criteria for article selection were defined a priori to avoid personal bias. In all, 106 articles were found in both the databases as well as the previously collected papers, 34 of which contained reports of wild plants specifically used for flavoring (excluding liqueurs and herbal teas). No data concerning plant lore are available for three regions (Veneto, Val D’Aosta and Trentino-Alto Adige), while no data about wild plants used as flavoring are obtainable from five regions (Puglia, Marche, Umbria, Molise and Lazio).

### 5.3. Data Analysis

Based on the results obtained, I set up a checklist including taxon, family, vernacular names, plant part(s) used, Italian regions for which the use of the species is reported, bibliographic citations, and therapeutic uses. The nomenclature follows the Plant List Database [99]. Families are organized according to APG IV [100] for angiosperms. When helpful, due to the recent changes in nomenclature, synonyms are reported in parentheses.

With a view to assessing the importance of herbs and spices in the study area, I used the Ethnobotanicity Index (EI), sensu Portères [101], which is the ratio, expressed as a percentage, between the number of plants used and total number of plants that constitute the flora of each region.

## Figures and Tables

**Figure 1 plants-10-00563-f001:**
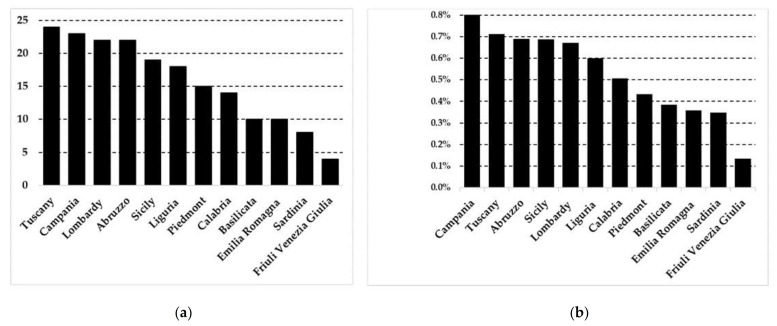
Number of species used per Italian regions (**a**) and EI for each region (**b**).

**Figure 2 plants-10-00563-f002:**
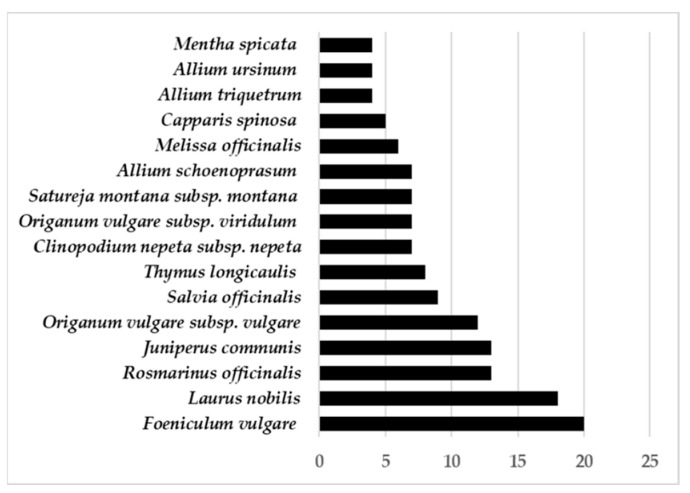
Most cited species in all Italian regions.

**Table 1 plants-10-00563-t001:** Wild herbs and spices traditionally used in Italy (Abr—Abruzzo; Bas—Basilicata; Cal—Calabria; Cam—Campania; Emr—Emilia Romagna; Fvg—Friuli Venezia Giulia; Lig—Liguria; Lom—Lombardy; Mol—Molise; Pie—Piedmont; Pug—Puglia; Sar—Sardinia; Sic—Sicily; Tus—Tuscany).

Species	Family	Vernacular Name	Part Used	Culinary Uses	Region	References	Therapeutic Uses
*Achillea millefolium* L.	Asteraceae	Tanéda	Flowers	Soup flavoring	Pie	[25]	N/A
*Achillea moschata Wulfen*	Asteraceae	Tanéda matta	Flowers	To flavor salt, pizzoccheri, risotto, salmì, semifreddo and sorbet.	Lom	[23]	N/A
*Allium lusitanicum* Lam.	Amaryllidaceae	Aij mat	Bulbs, leaves	Flavoring in several dishes	Lom	[23]	Blood pressure regulator.
*Allium neapolitanum* Cirillo	Amaryllidaceae	Agghie	Bulbs, leaves	Raw or cooked as herb	Cam	[26]	Blood pressure regulator.
*Allium nigrum* L.	Amaryllidaceae			Added to the tomato sauce	Sic	[22]	N/A
*Allium roseum* L.	Amaryllidaceae	Aglio selvatico, agghie	Bulbs, leaves	Raw or cooked in salads and soups	Abr, Cam, Tus	[16,26,27]	For gastrointestinal disorders.
*Allium schoenoprasum* L.	Amaryllidaceae	Erba cipollina, ĆiĨón, aiet, poureto	Leaves	To flavor rice, pasta, salads, soups, fritters and risotto	Emr, Fvg, Pie, Lom, Tus	[23,25,27,28,29,30,31]	Anthelmintic, diuretic, disinfectant, blood pressure regulator.
*Allium subhirsutum* L.	Amaryllidaceae		Leaves	To flavor salads	Sar	[32]	N/A
*Allium triquetrum* L.	Amaryllidaceae	Agghie, àggiu	Bulbs, leaves	To flavor potato cake, fritters, salads and soups	Cam, Lig, Sar; Tus	[26,27,32,33]	Anthelmintic, diuretic, disinfectant, blood pressure regulator.
*Allium ursinum* L.	Amaryllidaceae	Aglio ursino, agliustro, aglio selvatico	Bulbs, leaves	Flavoring in soups	Abr, Cam, Emr	[16,27,34]	Anthelmintic, diuretic, disinfectant, blood pressure regulator.
*Allium vineale* L.	Amaryllidaceae	Aglio pippolino	Bulbs, leaves	Flavoring soups, salads and sauces	Tus, Pie	[25,27,30]	N/A
*Anethum graveolens* L.	Apiaceae	F’nucchiastr	Leaves	Spice for sausages and Easter cakes	Bas, Mol	[35,36]	N/A
*Armoracia rusticana* G. Gaertn., B. Mey. and Scherb.	Brassicaceae	Cren, hrne	Roots	As spicy flavoring in sauces and soups	Fvg	[31]	N/A
*Artemisia vulgaris* L.	Asteraceae	Arsemize	Leaves	Flavoring soups and omelettes	Pie	[29]	N/A
*Brassica nigra* (L.) W. D. J. Koch	Brassicaceae	Senape	Seeds	Aromatic herb	Abr	[16]	N/A
*Calamintha sylvatica* L.	Lamiaceae		Leaves	Condiment.	Bas	[37]	N/A
*Capparis spinosa* L. (incl. *C. ovata* Desf.)	Capparaceae	Chiapparo, chiapparu	Buds	Flavoring for salads, pasta, meat, fish, sauces, pasta and garnishes to add a pungent spicy flavor.	Cal, Cam, Sic, Tus	[22,27,38,39,40]	Digestive and aperient.
*Carum carvi* L.	Apiaceae	Choré, chiréi, chimmel	Seeds	Flavoring (especially meat and poultry, particularly rabbit, or sarass, a local fermented and spiced ricottacheese). Flavoring in bread.	Lom, Pie	[23,28,29]	Carminative, digestive, galactagogue.
*Clinopodium nepeta* (L.) Kuntze subsp. *nepeta* (= *Calamintha nepeta* (L.) Savi.)	Lamiaceae	Anipeta, nepetella	Leaves	In soups and to aromatize artichokes, tomatoes, zucchini and chickpea salads, or as flavoring for fried or preserved in oil mushrooms.	Cal, Cam, Sic, Tus	[22,26,27,30,34,38,41]	Against stomach ache.
*Crataegus monogyna* Jacq.	Rosaceae	Biancospino	Fruits	As spice.	Cam	[34]	N/A
*Crithmum maritimum* L.	Apiaceae	Finucchieddu maritimum	Leaves	As flavoring.	Sic	[42]	N/A
*Cydonia oblonga L.*	Rosaceae		Fruits	Flavoring in the *sapa de ficu murisca*	Sar	[32]	N/A
*Daucus carota* L. subsp. *carota*	Apiaceae		Leaves	To flavor game meat.	Sar	[32]	N/A
*Foeniculum vulgare* Miller	Apiaceae	Finocchio selvatico, aniedd, fnucchio, finucc, fenùcciu, finocchiu sàrvaticum	Leaves; seeds; fruits; young stems	Fruits to flavor bread, dried figs and pickled olives, and sausages. Finely chopped leaves as flavoring in soups, omelettes, meat and fish dishes	Abr, Bas, Cal, Cam, Emr, Lig, Pug, Sar, Sic, Tus	[16,21,22,24,25,27,30,32,33,34,37,38,39,40,41,43,44,45,46]	Digestive, carminative, gastrointestinal disorders.
*Gymnadenia rhellicani (Teppner and E.Klein) Teppner and E.Klein (=Nigritella rhellicani* Teppner and E. Klein; *N. nigra* (L.) Rchb.)	Orchidaceae	Vaniglione, moréti	Flowers	Flavoring in sweets.	Lom	[17,23]	N/A
*Helichrysum italicum* (Roth) Don	Asteraceae	Rosmarina sarvaggia	Leaves	As spice.	Sic	[22]	N/A
*Juniperus communis* L.	Cupressaceae	Ginepro, zenéivu, zenéivru, giùpp, anèbri, genebbolo, zinebro, gënébbre, chais	Fruits	To flavor various dishes, and in particular those based on game (e.g., boar), roasted meat and fish	Abr, Cam, Emr, Lig, Lom, Mol, Tus, Pie	[16,17,23,25,27,28,30,33,34,36,43]	Against cough and sore throat, as diuretic and digestive.
*Juniperus oxycedrus* L.	Cupressaceae	Ginepro, zenéivu, zenéivru	Fruits		Abr, Lig	[16,33]	N/A
*Juniperus sabina* L.	Cupressaceae	Ginepro	Fruits		Abr	[16]	N/A
*Laurus nobilis* L.	Lauraceae	Alloro, l’uro, eufoggiu, osì, orbàga, lauru, addauru, orbaco, loriè	Leaves	To season boiled chestnuts, roast meats, jellied pork, codfish, sausages, chili peppers and dried figs. Much appreciated as a condiment is the extra virgin olive oil in which rosemary twigs and leaves are left to macerate [43].	Abr, Bas, Cal; Cam, Emr, Lig, Lom, Sar, Sic, Tus, Pie	[16,17,21,22,23,27,29,33,36,37,38,39,40,43,44,45,46,47,48]	Digestive, gastrointestinal disorders.
*Lavandula angustifolia* Miller	Lamiaceae	Steccadò	Flowers	For flavoring salads and meat.	Pie	[25]	N/A
*Melissa officinalis* L.	Lamiaceae	Melissa, cedroncella	Leaves	For flavoring salads and omelettes.	Abr, Cal, Cam, Emr, Tus	[16,27,34,43,49]	Spasmolytic.
*Mentha arvensis* L.	Lamiaceae	Menta	Leaves	Aromatic herb.	Abr, Cal	[16,38]	N/A
*Mentha longifolia* L.	Lamiaceae	Menta, mëntatre	Leaves	For flavoring salads and omelettes.	Cam, Fvg, Pie	[26,29,31]	N/A
*Mentha pulegium* L.	Lamiaceae	Menta	Leaves	In sauces and fish dishes.	Abr, Cal, Cam, Sic	[16,22,26,45,50]	N/A
*Mentha spicata* L.	Lamiaceae	Menta	Leaves	Flavoring for pancakes, sauces and stewed broad beans.	Abr, Bas, Cam, Sic, Pie	[16,22,26,28,37]	Digestive.
*Mentha* spp.	Lamiaceae	Menta	Leaves	Flavoring of potato ravioli and vegetable pies; fritters.	Lig, Pie	[25,43]	N/A
*Mentha suaevolens* Ehrh.	Lamiaceae	Menta, mentastra	Leaves	To aromatize grilled food, sauces, risotto, soups and salads.	Abr, Sic, Tus	[16,22,27]	Digestive.
*Mentha suaveolens* Ehrh. ssp. *insularis* (Req.) Greuter	Apiaceae		Leaves	To flavor a kind of black pudding.	Sar	[32]	N/A
*Mentha* x *piperita* L.	Lamiaceae	Menta piperita, mintaster	Leaves	Flavoring in salads, soups and risotto.	Tus	[27]	Digestive.
*Mercurialis annua* L.	Euphorbiaceae	Mercurella	Leaves	To aromatize codfish.	Tus	[27]	Aperient.
*Myrtus communis* L.	Myrtaceae	Murtidda, mortedda, mortella, murtuèlla	Leaves	To flavor game meat.	Cal, Lig, Sar	[33,48,50]	Gastrointestinal disorders.
*Nepeta cataria* L.	Lamiaceae	Mentastro	Leaves	Aromatizer for goat meat.	Cam	[51]	N/A
*Oenanthe pimpinelloides* L.	Apiaceae	Prezzemolo selvatico	Leaves	To aromatize boiled chestnuts, pig liver boiled.	Tus	[30]	N/A
*Olea europaea* L.	Oleaceae	Olivo, uivu	Leaves	To flavor meat (rabbit, boar, goat).	Cal, Lig	[33,38]	Blood pressure regulator.
*Origanum vulgare L.* subsp. *viridulum* (Martrin-Donos) Nyman (=*Origanum heracleoticum* L.)	Lamiaceae	Rigono, arìgano, harigana janca, rinu	Leaves	To flavor poultry, fish, and other dishes; as a spice in tomato-based sauces or salads.	Bas, Cam, Pie, Sic	[21,22,25,34,37,41,45,46,51]	Appetizer, carminative, depurative, digestive.
*Origanum vulgare* L. subsp. *vulgare*	Lamiaceae	Origano, orècano, arregano, harigana rossa, cornabuggia, orighen, regano	Leaves	To flavor poultry, fish, and other dishes; as a spice in tomato-based sauces or salads.	Abr, Cal, Cam, Lig, Lom, Tus, Pie	[16,22,23,27,29,33,38,43,44,50,51]	Appetizer, carminative, depurative, digestive, sedative.
*Papaver rhoeas* L.	Papaveraceae	Rosolaccio	Seeds	Flavoring for bread.	Tus	[27]	N/A
*Phlomis fruticosa* L.	Lamiaceae	Sarvia sarvaggia	Leaves	To aromatize meat and pasta flavoring.	Sic	[22]	N/A
*Pimpinella anisoides* V. Brig.	Apiaceae	Ciminu	Leaves	Flavoring for biscuits and taralli.	Cal	[38,49,50]	N/A
*Pimpinella anisum* L.	Apiaceae	Anice, anici, anèsc	Seeds	As flavoring in bread, sweet and dishes of meat.	Abr, Lom	[16,23]	Carminative, digestive.
*Pinus mugo* Turra	Pinaceae	Mügh	Buds	Flavoring dishes of fish and meat.	Lom	[17]	N/A
*Raphanus raphanistrum* L. subsp. *raphanistrum*	Brassicaceae	Rafano	Roots	To season spaghetti.	Cam	[41]	N/A
*Rosmarinus officinalis* L. (= *Salvia rosmarinus* Spenn.)	Lamiaceae	Rosmarino, rosamarina, rumaìn, ošmarín, rrosmarinu, tresmarino.	Leaves	Flavoring for meat roasts, potatoes, soups, bread and focaccia.	Abr, Bas, Cal, Cam, Emr, Lig, Lom, Sic, Tus	[16,22,23,26,27,33,37,38,43,44,45,46,47]	Digestive.
*Rubus idaeus* L.	Rosaceae	Mora	Fruits	Flavoring in dishes of meat.	Lom	[23]	Mild laxative.
*Ruta chalepensis* (L.) Savi	Rutaceae	Ruta	Leaves	Flavor for olive brine.	Lig	[43]	N/A
*Salvia glutinosa* L.	Lamiaceae	Salvia selvatica	Leaves	To flavor many dishes: soups, roasts, liver, rolls, beans, tortellini, sauces, etc.	Abr	[16]	Digestive
*Salvia officinalis* L.	Lamiaceae	Salvia, sarvia, addori ri costa	Leaves	To flavor many dishes: soups, roasts, liver, rolls, beans, tortellini, sauces, etc.	Abr, Bas, Cal, Cam, Lig, Lom, Sic, Tus	[16,22,23,27,37,38,43,49,50,51]	Digestive.
*Salvia pratensis* L.	Lamiaceae	Salvia selvatica, erba da osei	Leaves	To flavor many dishes: soups, roasts, liver, rolls, beans, tortellini, sauces, etc.	Abr, Emr, Lom	[16,17,27]	Digestive.
*Salvia sclarea* L.	Lamiaceae	Erva muscatiddara	Leaves	To aromatize various main dishes.	Sic	[22]	N/A
*Salvia verbenaca* L.	Lamiaceae	Cavolo moro	Leaves	To season salads, risotto and beans soups.	Tus	[27]	Digestive.
*Sambucus nigra* L.	Adoxaceae	Pepe di maio, sambucu, sambùch, savùcu	Fruits, flowers	Flavoring for bread, batters.	Cal, Lom, Sic	[23,38,40]	N/A
*Sanguisorba minor* L.	Rosaceae	Salvastrella	Leaves	To season soups and salads.	Tus	[27]	Astringent.
*Satureja hortensis* L. (Lamiaceae)	Lamiaceae	Santoreggia	Leaves	Flavoring.	Emr	[27]	N/A
*Satureja montana* L. subsp. *montana*	Lamiaceae	Sarapuddu, harihanedda, izzòppu, trombo	Leaves	Aromatizer, for goat meat, omelettes and risotto.	Bas, Cam, Lig, Pie, Tus	[21,25,26,27,33,43,51]	N/A
*Tanacetum vulgare* L.	Asteraceae	Archebüse	Leaves	Flavoring in omelettes and soups.	Pie	[29]	N/A
*Taraxacum campylodes* G.E.Haglund.	Asteraceae	Mourpoursin	Leaves	Pickled in brine and used as flavoring.	Pie	[29]	N/A
*Teucrium polium* L. subsp. *capitatum* (L.) Arcang.	Lamiaceae		Leaves	Flavoring.	Bas	[21]	N/A
*Thymbra capitata* (L.) Cav. (=*Thymus capitatus* (L.) Hoffmanns. and Link)	Lamiaceae	Tùmmaru	Leaves	To aromatize bread, fish and meat dishes.	Cal, Sic	[22,42,49]	N/A
*Thymus longicaulis* C. Presl (incl. *T. serpyllum* L.)	Lamiaceae	Timo, peverina, pepolino, serpoul	Leaves	As a condiment for omelettes, meatballs, flans, sauces, stuffed with meat, soups, snails, etc.	Abr, Cam, Lig, Tus, Pie	[16,25,27,28,29,34,43,44]	Digestive.
*Thymus polytrichus* Kern. ex Borbás	Lamiaceae	Érbapeverína	Leaves		Lom	[23]	Digestive.
*Thymus pulegioides* L.	Lamiaceae	Timo, retümmu, pepolino	Leaves	As *Th. longicaulis*.	Abr, Lig, Lom, Tus	[16,23,27,33]	Digestive.
*Thymus spinulosus* Ten.	Lamiaceae	Timo	Leaves	Aromatic herb.	Sic	[22]	N/A
*Thymus* spp.	Lamiaceae	Timo	Leaves	Aromatic herb.	Cal, Emr	[27,43]	N/A
*Thymus vulgaris* L.	Lamiaceae	Timo, retümmu, pepolino	Leaves	As *Th. longicaulis*.	Abr, Lig, Tus	[16,27,33]	N/A
*Trifolium medium* L.	Fabaceae	Trefòl	Flowers	As flavoring in cakes.	Lom	[23]	N/A
*Trifolium pratense* L.	Fabaceae	Trefòl	Flowers	As flavoring in cakes.	Lom	[23]	N/A
*Ulmus minor* L.	Ulmaceae	Olmo	Fruits	As flavoring in salads	Tus	[27]	N/A

## Data Availability

All the relevant data used for the paper can be found in Table 1.

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
