# Peer review of "Wild Plants Used as Herbs and Spices in Italy: An Ethnobotanical Review"

_plants, 2021, doi:10.3390/plants10030563_

Round 1

Author Response

Thank you for comments and suggestions that greatly improved my manuscript. All corrections have been accepted and incorporated into the manuscript.

Here attached please a point by point response 

Best regards

Reviewer 2 Report

Re: Manuscript Plants - 1150913

A. In the response to the reviewer, the author states that some suggestions have been incorporated in the text but in fact, they are not present in the corrected version of the article i.e.:

  • addition “N/A” or “-“ in places where data are not available;
  • Artemisia vulgaris – should be in italics;
  • Foeniculum vulgare – fruits are not included in the “Part used”;
  • Salmi is still in table 1;
  • A. lusitanicnum -flavoring in several dishes, examples are needed;
  • Ulmus minor, Olmo shifted;
  • Fig. 2 Taking into account the order of descriptions in the text, the first one on the graph should be Foeniculum vulgare;
  • Some “culinary uses” are still in species descriptions e.g. sage, lemon balm;
  • Some references still need revision; e.g. no 4 J. Ethnobiol. Ethnomedicine (with dots); no 15 J Ethnobiol Ethnomedicine (without dots); I suggest paying more attention to details to avoid the impression of carelessness.

B. This is a review article, so I suggest the author replace headings 'Results’ and ‘Discussion’ with those that refer strictly to the content of these sections. Citations of references in Results are not a standard practice.

C. In the case of the listing of the aims of this study, I meant the continuous, following form: "The aims of the study were to (a) document plant lore regarding wild herbs and species in Italy; (b) identify the wild …; (c) etc. "

Author Response

(The authors gave the same response as above.)

Round 2

Reviewer 1 Report

Dear author,

Thank you for addressing most of the issues raised. Nevertheless, you still need to provide more context regarding the ethnobotanicity index. From a simple Google Scholar search, there are at least a dozen papers that you could refer to, since they deal with this index, many of which focusing in the Mediterranean and in Italy, even though not specifically for herbs and spices, but e.g., for aromatical and medicinal plants. Finally, please add numbers to the different subsections of the Materials and Methods section.

Just a few such studies from Google Scholar:

Gras, A., Hidalgo, O., D’Ambrosio, U., Parada, M., Garnatje, T., & Vallès, J. (2021). The Role of Botanical Families in Medicinal Ethnobotany: A Phylogenetic Perspective. Plants10(1), 163.

Vitalini, S., Iriti, M., Puricelli, C., Ciuchi, D., Segale, A., & Fico, G. (2013). Traditional knowledge on medicinal and food plants used in Val San Giacomo (Sondrio, Italy)—An alpine ethnobotanical study. Journal of Ethnopharmacology145(2), 517-529.

Guarrera, P. M., Lucchese, F., & Medori, S. (2008). Ethnophytotherapeutical research in the high Molise region (Central-Southern Italy). Journal of Ethnobiology and Ethnomedicine4(1), 1-11.

Neves, J. M., Matos, C., Moutinho, C., Queiroz, G., & Gomes, L. R. (2009). Ethnopharmacological notes about ancient uses of medicinal plants in Trás-os-Montes (northern of Portugal). Journal of Ethnopharmacology124(2), 270-283.

Camejo-Rodrigues, J., Ascensao, L., Bonet, M. À., & Valles, J. (2003). An ethnobotanical study of medicinal and aromatic plants in the Natural Park of “Serra de São Mamede”(Portugal). Journal of ethnopharmacology89(2-3), 199-209.

Cornara, L., La Rocca, A., Terrizzano, L., Dente, F., & Mariotti, M. G. (2014). Ethnobotanical and phytomedical knowledge in the North-Western Ligurian Alps. Journal of ethnopharmacology155(1), 463-484.

Author Response

Thank you very much  for the important contribution to the present manuscript. I have consulted the papers you suggested to me.

Consequently lines 85-87 has been rephrased as follows:

The Ethnobotanicity Index values are significantly lower than those of other Italian regions (range 5.4-11%) or Iberian peninsula (range 8.8-27.9%) that are calculated on medicinal, cosmetic, veterinary, and food species [17,102,103]. Although it is not possible to make a comparison with the same use category, results suggested that the knowledge of wild plants used as flavoring is still consolidated in the above mentioned regions that are also among the most species-rich in Italy [28].

The following references have been added

  1. Camejo-Rodrigues, J., Ascensao, L., Bonet, M. À., & Valles, J. (2003). An ethnobotanical study of medicinal and aromatic plants in the Natural Park of “Serra de São Mamede”(Portugal). Journal of ethnopharmacology, 89(2-3), 199-209.
  2. Guarrera, P. M., Lucchese, F., & Medori, S. (2008). Ethnophytotherapeutical research in the high Molise region (Cen-tral-Southern Italy). Journal of Ethnobiology and Ethnomedicine, 4(1), 1-11.

Finally the different subsections of the Materials and Methods section have been numbered.

Round 3

Reviewer 1 Report

Dear author,

Thank you for responses to the issues raised. I have no further comments to make.